



# 1 **Graphic design and scientific research: the INGV experience**

Daniela Riposati[1], Giuliana D'Addezio[2], Francesca Di Laura[1], Valeria Misiti[2] and Patrizia Battelli[3]
[1] Istituto Nazionale di Geofisica e Vulcanologia, AC, Via di Vigna Murata 605, 00143 Rome, Italy
[2] Istituto Nazionale di Geofisica e Vulcanologia, Sezione di Sismologia e Tettonofisica, Roma1, Via di Vigna Murata
605, 00143 Rome, Italy
[3] Istituto Nazionale di Geofisica e Vulcanologia, ONT, Via di Vigna Murata 605, 00143 Rome, Italy
*Correspondence to:* Giuliana D'Addezio (giuliana.daddezio@ingv.it), Via di Vigna Murata 605, 00143 Rome, Italy

## 9 1. Abstract

Part of the INGV activity is focused on the production of resources concerning Educational and Outreach projects on
Geophysics and natural hazard topics. The forefront results of research activity, in fact, are periodically transferred to
the public through an intense and comprehensive plan of scientific dissemination. In the past 15 years, graphic and
visual communication has become an essential point of reference supporting institutional and research activities.
Positive experiences are the result of a strict relationship between graphic design and scientific research, in particular
the process concerning the collaborative work between designers and researchers. In projects such as the realization of
museum exhibition or the production of illustrative brochures, generally designed for broad-spectrum public, the goal is
to make easier the understanding and to support the scientific message, making concepts enjoyable and fruitful through
the emotional involvement that visual image can arouse. The graphics and editorial products, through composition of
signs and images by using different tools (colors, form, lettering) on different media (print, video, web), link to create a
strong identity "INGV style", in order to make them easily recognizable in Educational and Outreach projects. A
project product package might include a logo or other artwork, organized text and pure design elements such as shapes
and colour, which unify the piece. Colour is used not only to help the "logo" stand out from the international overview,
but in our case to have a unifying outcome across all the INGV sections. A recent and stimulating experience has been
the collaboration between INGV project design and its reference scientific community in order to create edu-games,
products specifically designed for scientific dissemination. The edu-games have been designed to be an efficient
combination of educational content and playful communicative aspects, with the aim therefore to learn while having
fun.

## 29 2. INTRODUCTION

The Istituto Nazionale di Geofisica e Vulcanologia (INGV) is one of the largest European research institutes of
geophysics, geochemistry, seismology and volcanology. INGV pays special attention to Education and Outreach
projects through publications for general public and/or devoted to schools, scientific divulgative exhibitions and
dedicated Internet pages.
The Laboratorio Grafica e Immagini (hereafeter Laboratorio) is the reference structure of the INGV for the graphic and
visual communication, supporting institutional and research activities. Since 2001, the Laboratorio has become an
important partner of the INGV scientific community for the realization of a reference point for the creation of visual
design products.
In the consideration of increasingly emphasis on the value of graphics in grant proposal (National Science foundation,
2004), the Laboratorio provides advanced graphics support for the editorial lines of the main European research
projects infrastructures and partnerships involving INGV. Among these we describe the case of SPACE EARTH,



EMSO-ERIC, TSUMAPS-NEAM, IAPG, AGITHAR, FIERI and SaveMedCoast.
The laboratorio's support for editorial productions is also combined with the development and production of new web
layouts dedicated to the representation of issues relating to Earth Sciences and their dissemination to the general public.
In this regard, we present the ScienzaInsieme project. As already mentioned particular attention was given to *ad hoc*
production designed for education and outreach projects. Over the years the activities of the Laboratorio Grafica e
Immagini have turned to the conception, design and realization of the popular character editorial materials, in both
ordinary and institutional purposes at exhibitions, demonstrations and special events. Exhibitions and specific
installations that has designed and implemented for exhibits about science and scientific communication (D'Addezio et
al, 2014; D'Addezio et al. 2015; Rubbia et al. 2015). In this purpose we present the multi-year participation of the
INGV at the Festival della Scienza di Genova, an unmissable appointment for all science enthusiasts, professionals and
not only and the New Space Economy European ExpoForum, held in Rome in December 2019.

## 3.  THE PARTNERSHIP BETWEEN GRAPHIC DESIGN AND SCIENTIFIC RESEARCH
Main Laboratorio's tasks focused on finding the right relationship and cohesion between interpretation of scientists
work through the use of graphic design using proper images and products. The goal is to elaborate appropriate solutions
to transfer purely scientific information addressing the messages not only to the pertinent scientific community but also
to general public, looking for the right compromise between visual design of the graphic work and the one of the
scientists. But what do we mean by graphic design, what is graphic design and why it is so important for graphic
composition in scientific communication? Graphic design is the process of visual communication, and problem-solving
through the correct use of typography, space, image and color. Graphic attracts viewer and graphic designers use
various methods to combine words, symbols, and images to create a visual representation of ideas and messages. The
importance of communicate complex ideas with clarity, precision and efficiency avoiding ambiguity and confusion, was
initially developed in the field of data visualization and information design (Tufte, 1983). In these themes, graphical
excellence is what gives to the viewer the greatest number of ideas in the shortest time, with the least ink in the smallest
space (Tufte, 1983). A graphic designer may use a combination of typography, visual arts and page layout techniques to
produce a final result. Common uses of graphic design may include corporate identity, publications, posters, website
graphics and elements, product packaging. For example, a product package might include a logo or other artwork,
organized text and pure design elements such as images, shapes and color which unify the piece. Composition is one of
the most important features of graphic design, especially when using pre-existing materials or various elements.
In visual scientific communication, common uses include the development of the institutional identity of research
bodies and scientific projects, the layout of publications, posters for sector conferences, web communication,  design of
the entire visual communication of exhibitions aimed at the general public (from logo to panels, up to gadgets and web
promotion).
In any case, the opted of graph is to operate a constant mediation between scientific concept to be represented, and
visual form that can represent it more clearly.

## 4.  THE INGV STYLE
The exposed approach was used for INGV productions endorsing the creation of a INGV identity: an image strongly
characterized in terms of style, an INGV brand productions highly recognized in the scientific community. This
"identity" has played an important role on different products, addressed to a target audience or for contents. The most
important product was the restyling of the INGV logo.




### 4.1 The INGV new logo

Considering the almost thirty-years life of the INGV logo, we proposed and realized its restyling mainly modernizing
the image of the previous one. The INGV logo consists of two parts: one graphic and the other textual. For the text,
assuming an idea of modernity, the Arial bold black 100%, has been replaced by a more actual and clean DIN Pro Bold
Consended gray 90%, used in small/caps. The graphic part represents a schematic and sectioned reproduction of the
Earth, has lost the elements that strongly characterized it, the serrated lines simulating parallel and meridians. The
element has therefore returned to a simpler sphericity, accentuated by the chromatic nuance. This solution has already
been declined in the production of new editorial projects and on the occasion of national and international events as we
will see later (see Figure 1).

### 4.2 The INGV anniversary

One of the most important events that involved the Laboratorio was the INGV twentieth anniversary in 2019. We
studied the logo for the event, starting from the analysis of the keywords of the title: - twenty years, - geosciences -
travel / future. Analyzing the visual communication of the publications dedicated to geosciences, it was found that
frequently the most used image is stratigraphy and secondly mountains. For this purpose, we have developed a graphic
project very tied to the institutional image to which we have added a horizontal sign that reminds the stratigraphy and
visually represents the separation between before and after, above and below and remembers an arrow indicating the
movement towards the future, obtaining a strong but not didactic "symbolic" element. The yellow color evokes the
preciousness of gold on special occasions. We then developed the coordinated image, adapting it to the various
materials provided for the event (internal signage, presentations, web promotion, gadgets) so that it lent itself to
versatile configurations, also with respect to problems related to reproduction and spaces available, while maintaining
visual integrity intact (Figure 2).

## 5.  CORPORATE IDENTITY FOR RESEARCH PROJECT

### 5.1 SPACE EARTH www.spacearth.net

The Laboratorio realized the entire branding of the INGV spin-off company SPACE EARTH: a team of engineers,
physicists and geologists with a long experience in research and business management. The company aimed to add
value from the results of more than 60 years of experience on Space Earth designs and develops applications, software
and hardware products for the Aerospace, Maritime and Environment fields, in cooperation with major European and
Italian public and private organizations, universities and research centers. The Spacearth Technology logo was
conceived to graphically summarize the content of the message: "Space-Earth-Technology". For this purpose, familiar
forms have been used, such as two intersecting circles, to represent the Earth and the space that surrounds it at first, and
to a more in-depth reading of the relationship between SET project and INGV, the area that allowed the project to be
born and develop.
Chromatically, a single color was chosen that refers to the "iconic blue of the sky", to reaffirm the meaning and add an
emotional value to the pictogram. Even the chosen lettering is simple, linear, sans serif, just to reaffirm the modern and
technological aspect of the whole and to give the logo further immediacy, making it easy to decipher and therefore to
remember (Figure 3).

### 5.2 EMSO-ERIC www.emso.eu





The European Multidisciplinary Seafloor and water column Observatory (EMSO), aims to explore the oceans, to gain a
better understanding of phenomena happening within and below them, and to explain the critical role that these
phenomena play in the broader Earth systems. EMSO is a consortium of partners sharing in a common strategic
framework scientific facilities (data, instruments, computing and storage capacity). Formally it is a European Research
Infrastructure Consortium (ERIC), legal framework created for pan-European large-scale research infrastructures.
The contribution of the Laboratorio on the brand of this very important infrastructure was a "textual" intervention: the
acronym "ERIC" was inserted in the logo already developed.
On the textual insert, the same color nuance was used in the gestural element of the EMSO logo: this solution allowed
us to link the different parts with a simple but at the same time extremely effective interpolation in terms of image
return: it was possible to risk a "untying" between the various components that would certainly have weakened the
overall image. A whole series of products have therefore been designed and manufactured under the new brand EMSO-
ERIC (Figure 4; Dañobeitia, et al., 2019).

**5.3 TSUMAPS-NEAM Project www.tsumaps-neam.eu**
Tsunami risk assessments and warning systems need Probabilistic Tsunami Hazard Assessment (PTHA) as input and
reference. TSUMAPS-NEAM will develop the first homogeneous long-term PTHA for earthquake-induced tsunamis,
which is presently unavailable for the coastlines of the NEAM region (NE Atlantic, the Mediterranean, and connected
seas). TSUMAPS-NEAM will also promote an informed process of outreach, guidelines definition, and capacity-
building initiatives dedicated. The development of standardized PTHA products (hazard and probability curves, maps,
documentation, web-tools for their analysis) is the first step to include also tsunamis in multi-hazard risk assessments.
In designing the project logo we focused on some elements, deliberately moving away from the classic idea of graphic
representation of Tsunami, the wave. In fact, we believed that focusing on different elements allowed us to achieve a
more original and therefore highly recognizable creation: here is the stylization of the "hands" to signify the help that
the scientist's job can represent in the probabilistic study of tsunami hazard. The choice of colors was focused on how to
distinguish anthropic element from natural element, therefore a full orange was associated with human element and one
blue with the other. The fusion of the two colors in the intertwining of the hands which, as we have seen, wants to
represent cooperation, giving life to a transparency that increases the desired effect (Figure 5).

**5.4 International Association for promoting Geoethics (IAPG) www.geoethics.org**
IAPG is a multidisciplinary, scientific platform for widening the discussion and creating awareness about problems of
Geoethics and Ethics applied to Geosciences. IAPG promotes geoethics through international collaboration with
Associations and Institutions. The Laboratorio Grafica e Immagini has been supporting the IAPG activities for many
years.The great novelty of Geoethics had to start with a strong, recognizable image: for this reason, we started with
design of the logo, which would then be declined on all "new brand"products. We focused on the idea of interaction
between human activities and Earth system: the use of circular elements, their concentricity gave us the possibility of
creating a substantially spherical solution, with a core where patterns and textures were concentrated to represent social
diversity and stylized a point of intersection between Geosciences, Sociology, Philosophy and Economy. The products
made have been manifold (Figure 6).

**5.5 AGITHAR – Accelerating Global Science in Tsunami HAzard and Risk analysis www.agithar.uni-**
**hamburg.de**





AGITHAR - is a network created to improve, standardize, and promote tsunami research. We therefore concentrated on
a reinterpretation of tsunami wave, where the play of colors, shades and textures inserted, plays down the idea of
danger, which is instead treated by insertion of different colors both for lettering and for other graphics. The proposal is
extremely expendable given the box in which all graphic elements are contained and therefore allows its use that can
range from classic web use to the uses foreseen in the coordinated image (Figure 7).
**5.6 FIERI**
FIERI (forum for International cooperation among environmental research infrastructures) is an international open
platform for improving global, coordinated and long-term cooperation between Research Infrastructures and Networks
in environmental domain. What we thought in creating the logo was to highlight the connection aspect, a sort of
synapse that connects to Earth, forming a sort of global network. For colors we focused on a very green, modern and
bright idea, as well as for the used lettering (Figure 8).
**5.7 SAVEMEDCOAST - www.savemedcoasts.eu**
SAVEMEDCOASTS aims to respond to the need for people and assets prevention from natural disasters in the coastal
zones of the Mediterranean Sea, undergoing to increasing sea level rise due to climate change, coastal land subsidence,
tsunamis and storm surges impacts. The focus are coastal zones prone to sea level rise. The objective is the
stakeholder's preparation in facing the effects of these potential impacts. In these consideration, in the project logo
realization, we focused on wave graphics and anthropic element. We therefore extremely stylized and differentiated
them with the use of color. For lettering we used a sticks font, very squared accompanying it in the leaflet to another
font from another family but more versatile. There were many products made by the Laboratorio for this project. In this
regard, the declination of the logo was fundamental: finding the most suitable solution easily allowed us to impose the
recognition of the new "brand" on the scientific community of reference (Figure 9).
**6. WEB PRODUCTS: relating to Earth Sciences and their dissemination to the general public**
**6.1 SCIENZAINSIEME - www.scienzainsieme.it**
The ScienzaInsieme project faced with the need to create a common portal to multiple National Research Bodies and
Universities, with the aim of creating a system that becomes a lasting structure over time, through which advertise
scientific events dissemination. We have chosen to identify the evocative pictographs elements that represent both
science than sharing. Infinity is a very ancient symbol, used in different areas and whose birth is explained in
heterogeneous ways but all related to concept of quantity, time and space.
• The symbol of inverted eight - associated with alchemy, to Hermeticism and Gnosticism - as a variant of Ouroborus,
the snake or dragon that bites its tail, represents the theory of eternal return, the cyclical nature of all things. It is
attributable to all that can be represented through a cycle which, after reaching its end, starts again from the beginning,
to infinity. It was first represented in an ancient funeral text Egyptian, found in Pharaoh Tutankamun Tomb.
• Its origin in Roman times is attributable to the use of CI letters indicating very large, higher values to 1000.
• As a mathematical symbol, ($\infty$ - lemniscata), it was first adopted in 1655, to identify a very large number, just because
those two eyelets they can be endless paths.
• The analemma, which in astronomy indicates a particular geometric curve in the shape of eight, which describes the
position of the Sun in different days of the year, at the same time and in same location. A path that always begins and
ends in the same point thus representing "The eternal coming and going weather". Starting from the study of this



symbol in its perfect geometry, we chose to deform one of the two mirror parts to create two communicating sets,
through which the contents they mix to give shape to a new entity, in a virtuous circle and infinite of sharing and
creation (Figure 10).
**7.  EDITORIAL PRODUCTS: The interaction between graphic design and scientific production for the**
**reference scientific community and for scientific dissemination in a popular context.**
**7.1 The Laboratory for High-Pressures and High Temperatures of experimental geophysics and volcanology**
**annual Report**
In recent decades, the dizzying development of knowledge on technology and materials science has made it possible to
build tools capable of reproducing environmental conditions that control the dynamics of chemical-physical processes
inside and on the Earth's surface. Among these processes, those relating to seismicity and magmatism-volcanism are of
particular economic and social importance for the number of victims and the extent of damage they cause. In this
context, the Laboratory for High Pressures and High Temperatures of Experimental Geophysics and Volcanology
(HPHTLab) has developed at INGV. The Laboratorio then created the editorial graphic project of the Annual Report of
the HPHT Lab. The report is aimed essentially at an audience of professionals and at the editorial level represents an
excellent combination of graphic design and geoscience research, also intended for its more advanced sectors (Figure
221  11).
**7.2 School calendars**
A significant part of the work is focused on the achievements for scientific dissemination. For example, the celebration
of 10 years of a very successful initiative: the calendar dedicated to the primary schools designed to support and
integrate the outreach activities conducted for over fifteen years with the schools (D'Addezio, submitted). The graphic
design was aimed at producing an "object" that would gather the 10 calendars with a common target: The Planet Earth,
10 years with the Earth seen by the children (Figure 12).
**7.3  EDU GAMES**
In recent years, attention in themes of education  has focused on the production of scientific games, an efficient
combination of educational content and playful communicative aspects, with the aim therefore to learn while having
fun. Among these productions, Escape Volcano, Mareopoli and Cacth the Plate stand out for interest and public
success.
**7.3.1 Escape Volcano**
The game we present is devoted to transmit basic notion on volcanoes and its eruption types and also on environmental
and earthquake risks (Di Nezza et al., 2019; Misiti et al., 2019). Basically, the game is composed by plastic billboard
160*200cm which represents a volcano with its magmatic chamber. Small chambers, ten in total, are located along the
conduit up to crater. The goal of the game is to reach the crater before volcano eruption overcoming different tests. Four
pawns, that represent small volcanoes, are positioned in the magmatic chamber. Minimum 2 and maximum 4 teams are
supposed to play. To move from one chamber to the next one the players have to roll a dice. On the faces of the dice are
reported tests that players have to pass. The game has been thought and realized with some high school students in the
frame of Alternanza Scuola Lavoro project. The design and construction phase involved first of all:
- analysis of the idea developed by the students;



- study of target audience;
- and evaluation of problems related to production and practicality of use of instrument, in order to make it easy to
handle, easily transportable and reproducible even with simple and economic means.
The centerpiece of the game is a large format billboard, designed to allow at least 20-25 players to participate
simultaneously. As it was not a self-explanatory game, it was necessary to emphasize the visual aspect to enhance
emotional impact on participants. With this aim we have chosen to characterize the whole coordinate, from game board,
to various components such as: cards, assembly boards of the pieces, 3D pieces, dice and rules, with pastel colors and a
playful lines graphic, easily adaptable to youth target audience tastes.
The sinuous forms with which the volcano was represented refer directly to classical iconography, however deprived of
the didactic aspect and of any scientific reference, just to highlight the playful character of the instrument. We have
chosen to characterize the various parts of the game through icons, deliberately winking at the social ones, to seek a
familiar connection in visual baggage of today's kids, which would make involvement in the activity even more fluid.
Even the typographic choices have been oriented in this sense. The use of a calligraphic character (Princess Ivy) that
strongly connoted the visual aspect, in main titles of all the game components, is dictated by the need to create a
dominant visual element of entire project that conveys a sense of dynamism, of freedom, but also lightness. The
prerogative of calligraphic characters is precisely to have the graces, the ascendants and descendants very elaborate and
pronounced, which refer to handwriting trait and allow users to create a more "artistic and emotional" visual. This,
however, at the expense of readability, which in fact has been supported by the use of explanatory texts, of a simpler
(Rotis serif) font. Great weight was also given to the use of black color, which aimed to make style of the entire product
more adult in older children's eyes, in order to involve them without diminishing their age. Proposing a too childish
aspect could have created a preconception in adolescent participants, thus reducing effectiveness of message and ability
to receive information. (Figure 13).

**7.3.2 Mareopoli**
The game is inspired by the famous board game MONOPOLY. It will be realized in two formats, a big version in order
to be played in groups at recreational-scientific laboratories, and a small version as a gift for participants and as a take-
home message (Locritani et al., 2018). The game describes scientifically tides and historical theories on their origin
from the greek period till the end 18th century (Taramaschi, 2013). Many scientists have tried to understand and
interpret this phenomenon. Among the oldest the game quotes Aristotle and Eratosthenes, but also other eminent
seventeenth century scientists such as Galileo Galilei, up to the physicists who formulated modern theories as Newton
and Laplace. Finally, the game gives scientific information on cross-cutting issues related to tides as: renewable energy,
biodiversity and ecosystem conservation. This game is the result of continuous collaboration between researchers and
graphic designers: working together simplified scientific concepts and translated them into compelling and direct
images. The most relevant historic and scientific topics have been simplified into fundamental concepts, while
maintaining a common conceptual and stylistic line, and choosing two-dimensional drawings, although some shading is
used to introduce a sense of background, perspective or motion. Nevertheless, it has been attempted to keep drawings as
simple, plain and clear as possible in order to convey specific ideas in a more effective way. All illustrations have been
made in the Laboratorio with painting techniques (Figure 14).

**7.3.3 Catch the Plate**



Catch the plate is a game as simple as it is addicting. Children and young people, from 11 to 16 years old, will always
be able to play perfectly under the guidance of a conductor. The participants of the game must divide into teams made
up of a minimum of two players. The team with the youngest player will start. The team that starts, will roll the dice.
Each roll of die determines which card or token must be drawn and consequently the actions to be carried out combined
with each one, listed below:
1) *EARTHQUAKE CARD*
2) *VOLCANO CARD*
3) *TECTONIC PLATE*
The objective of the game is to place the largest number of tectonic plates, earthquakes and volcanoes to get the highest
score. The game is thought to teach children and people how the Earth moves, and how is made the Earth crust.
*EARTHQUAKE CARD*
A card with the earthquake epicenter drawn is delivered. The goal is to guess where to place the epicenter on the basis
of questions shown on cards. Placing correctly, you win 3 points. If the team requires an extra clue to guess it gets 2
points (in case of exact answer). You can also give your turn to the opposing team that in case of correct answer wins 1
point.
*VOLCANO CARD*
A volcano made of das is delivered. The goal is to guess where to place the volcano on the base of the application
shown on cards. By positioning correctly, you win 3 points. If the team requires an extra clue to guess it gets 2 points
(in case of exact answer). You can also give your hand to the opposing team that, in the case of a correct answer, wins 1
point.
*TECTONIC PLATES*
Main Earth plates are 15 in total. Players will have to draw a plate from a basket and place it correctly on the board. If
the team misses the plate, it is put back into play. Guessing the plate immediately wins 3 points (Figure 15).

### 310 7.3.4 Geo Trivial

The latest product created within the edu-games is the GEO-Trivial. As it is known, games have the power to ignite
imaginations and place you in someone else's shoes or situation, often forcing you into making decisions from
perspectives other than your own. This makes them potentially powerful tools for communication, through use in
outreach, disseminating research, in education at all levels, and as a method to train the public, practitioners and
decision makers in order to build environmental resilience.
By creating the Geo Trivial game we thought to revisit the classic Trivial, thus producing a scientific game, a tool to
learn more about the amazing world of geosciences by enjoying. This new game belongs to a INGV editorial project
dedicated to education and outreach (Figure 16, work in progress).

## 320 8. GRAPHIC DESIGN AND SCIENTIFIC RESEARCH

### 321 8.1 INTERACTIVE EXHIBITION

#### 322 8.1.1 Il pianeta dei cambiamenti: la tettonica delle placche: una teoria rivoluzionaria - Festival della Scienza di
Genova 2018
The exhibition aimed to tell the fundamental steps, discoveries and intuitions that provided intellectual and disciplinary
credibility to the Plate Tectonics Theory, one of the most important scientific acquisitions of the twentieth century. Its
enunciation followed a golden age for the discoveries of Earth Sciences, helped the scientific community to accept the



basic ideas underlying the drift of the continents, laying the foundations for a change in our perception of dynamics of
the planet. By bringing together results from various disciplines, the theory has unveiled the dynamics of our planet,
forever revolutionizing Earth Sciences.
The study of the logo has therefore focused on the Earth and its complexity: the geometric elements can remember a
puzzle, a puzzle that is composed and decomposed like Earth, a planet that is always on move.
The exhibition was set up at the prestigious premises of Palazzo Ducale in Genoa, which, precisely because of their
uniqueness and beauty, have therefore allowed a setting of great impact (Figure 17). The exhibition welcomed visitors
in a play of light and color and accompanied him throughout the journey. We have indeed chosen this key of
interpretation (light and color) to characterize the exhibition. The public success was remarkable for the whole
initiative.
The exhibition is also having an editorial following: The Exhibition Catalog is in fact released, which has become a real
book that tells a history of changes. On the one hand the changes of our planet, a living and constantly changing
environment; on the other, the changes in the way of thinking, seeing and explaining the world that, over two thousand
years, have guided man in understanding the mechanisms that govern the evolution of the Earth. In figures 17 and 18
we show the logo, some creations (panels, gadgtes and photos of the exhibition) including a visual summary of the
Catalog.

### 8.1.2 Attenti agli elementi – Festival della Scienza di Genova 2019

"Earthquakes: beware of the elements! Details that save lives", created for the Festival della Scienza di Genova 2019
and now itinerant with appointments scheduled in many cities of Italy (Grottaminarda, Varese, Milano, L'Aquila, etc.).
The aim of the exhibition is to illustrate good practices to prepare for earthquakes and increase citizens' awareness
on the Earth dynamic environment in constant evolution.
In an interactive journey, visitors will discover how the different elements that make up a building react to earthquake
shocks and what is the role of land on which our houses are built.
In the graphic  project for the exhibition we started from the choice of a vintage style, so that it would result in a
modern but not too minimalist appearance. The intent was to produce a "familiar and intimate" communication instead
of institutional, cold and authoritarian, so that the message was conveyed in an empathetic and welcoming way in order
to obtain a greater availability to transmitted concepts. The dominant color is orange, chosen as a compromise with red
which instead of evoking an emergency, is usually used to recall: cheerfulness, sociability, vitality and renewal. It
seemed perfect for this popular exhibition where dynamism is synonymous with awareness and action. The icon of the
house is the dominant element together with the crack in the ground and the chandelier that oscillates, now part of the
collective imagination related to earthquake risks related. The objective of the visual project was to produce a playful
visual communication that represented, together with the negative aspect of natural risk, also the positive aspect of
awareness of the structures and of the rules of behavior, which can save our lives and which are the focus of the
exhibition. In Figures 19 and 20 we show the main products made and some photos of the set-up.

### 8.1.3 NSE Spac Forum

INGV participated to the first NSE European Expoforum Italian edition, a point of reference for companies that
operates in Space sector, but also and above all for all those companies that orbit the New Space market: Universities,
SMEs, Research Centers, innovative companies. We have chosen to characterize our exhibition space by focusing
strongly on a visual element, an image of a volcanic eruption from space. This choice was then declined on the scientific





products, flyers, ad hoc created for the event (Figure 21 and 22).

**9. Conclusions**
Visual culture has become a prominent part of the cultural identity in the 21st century and consequently, is an important
tool with which to communicate science. On the other hand, visual material is typically treated as an add-on instead of
being an integrated part of the whole and there is a lack of identifying target audiences and refining visual elements for
them specifically (Rodríguez Estrada and Davis, 2014; Khoury et al., 2019). In our experience science communication
become more effective visual communications by integrating and incorporate elements of theory and practice from the
discipline of design. But if the wealth of these tools, especially considering online science communication, allows to
experiment with increasingly effective communication models, on the other hand the scientific image risks losing its
original explanatory function to adapt to technical requirements and aesthetic standards (Rigutto, 2017). To outlines the
importance of deep cohesion between graphic support and scientific message it is also the evaluation that usually
viewers tend to rely on preexisting levels of trust and peripheral cues, such as source attribution, to judge the credibility
of shown data (Li et al., 2018). The INGV experiences we present, between researchers, graphic designers, and other
visual communications highlight a great potential and a virtuous example of compromise between strictly
communicative needs and correctness of information which is the core of communicative and visual message. Finally,
we believe that this type of collaboration is a fundamental component in the dissemination of scientific information
towards the general public and in educational context.

**10. Author contribution**
Daniela Riposati is the Coordinator of the INGV Laboratorio Grafica e Immagini. Her contribution on this work have
been focused on writing, visual and content research, with the aim of creating a homogeneous and usable product.
Giuliana D'Addezio as Coordinator of INGV Laboratorio Attività con le scuole, cooperates closely with the
Laboratorio Grafica e Immagini. She provided research materials, ideas and inputs for discussion.
Francesca Di Laura is one of the fundamental components of the Laboratorio. Her contribution in drafting the paper
have been focused on the writing and general approach of the article.
Patrizia Battelli, new and recent entry into the Laboratorio, has contributed in drafting the paper with presence, valuable
advice and helping the general review.
Valeria Misiti helped as the scientific support for the edu-games produced by Laboratorio.

**11. Acknowledgements**
The Authors want to thanks all colleagues who have supported and sponsored over the years the activities. Thanks to
their support it has been able to achieve these results. In particular we want to thank Angela Chesi and Sabrina Palone,
two colleagues who have shared this trip with us for a few years.
The authors declare that they have no conflict of interest. Figures are from INGV publications and productions.

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



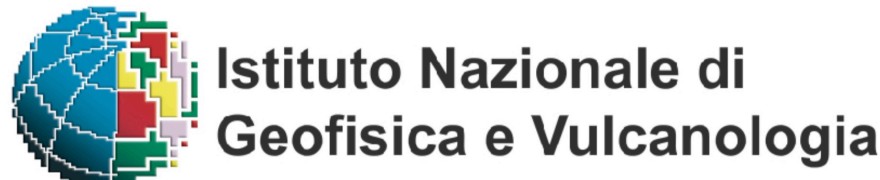

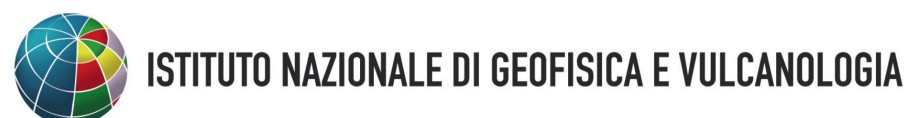

**Figure 1: Above the logo of the INGV dating back to 1986. Below the revisitation of 2018. Copyright Laboratorio Grafica e Immagini INGV.**




Figure 2: INGV Twentieth anniversary products. Copyright Laboratorio Grafica e Immagini INGV.







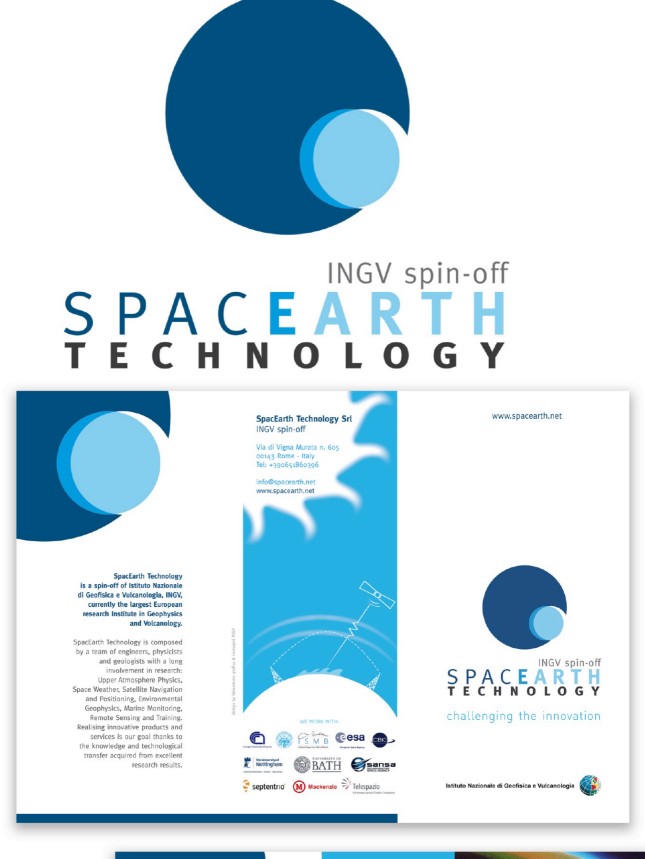

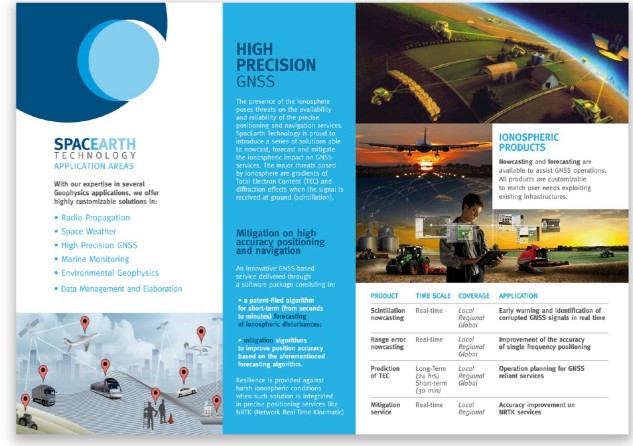

**Figure 3: The logo designed and created for the Space Earth spin-off and its declination in the project brochure. Copyright Laboratorio Grafica e Immagini INGV.**





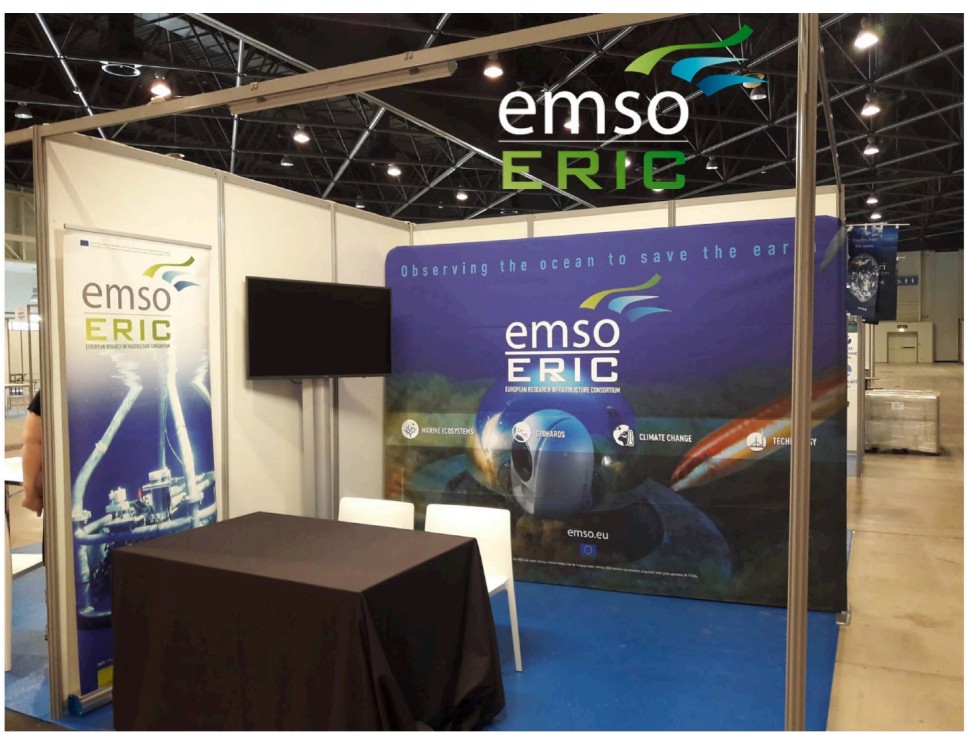

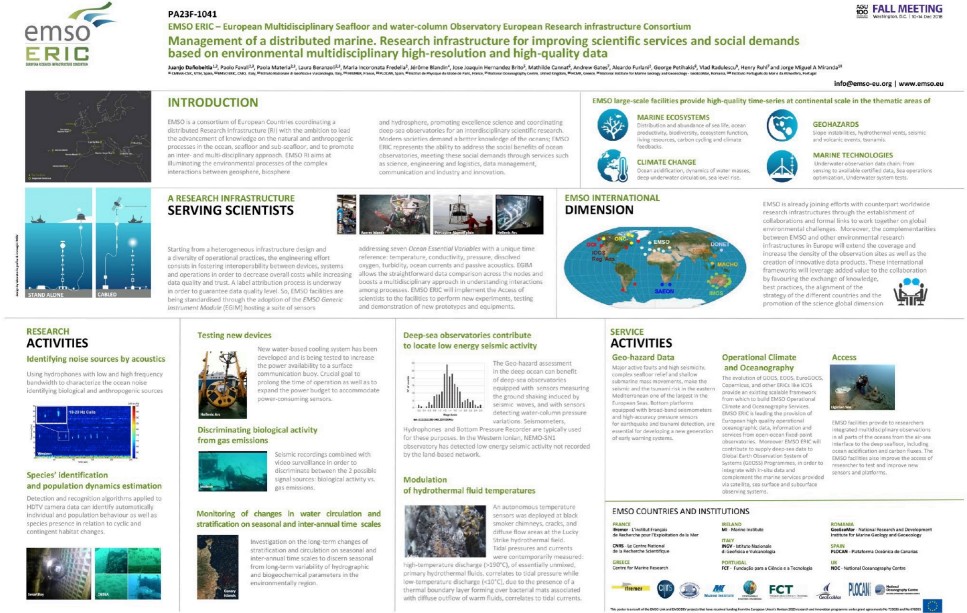

**Figure 4: The EMSO ERIC logo and its declination in some products (congress stands, totems, posters). Copyright Laboratorio Grafica e Immagini INGV.**





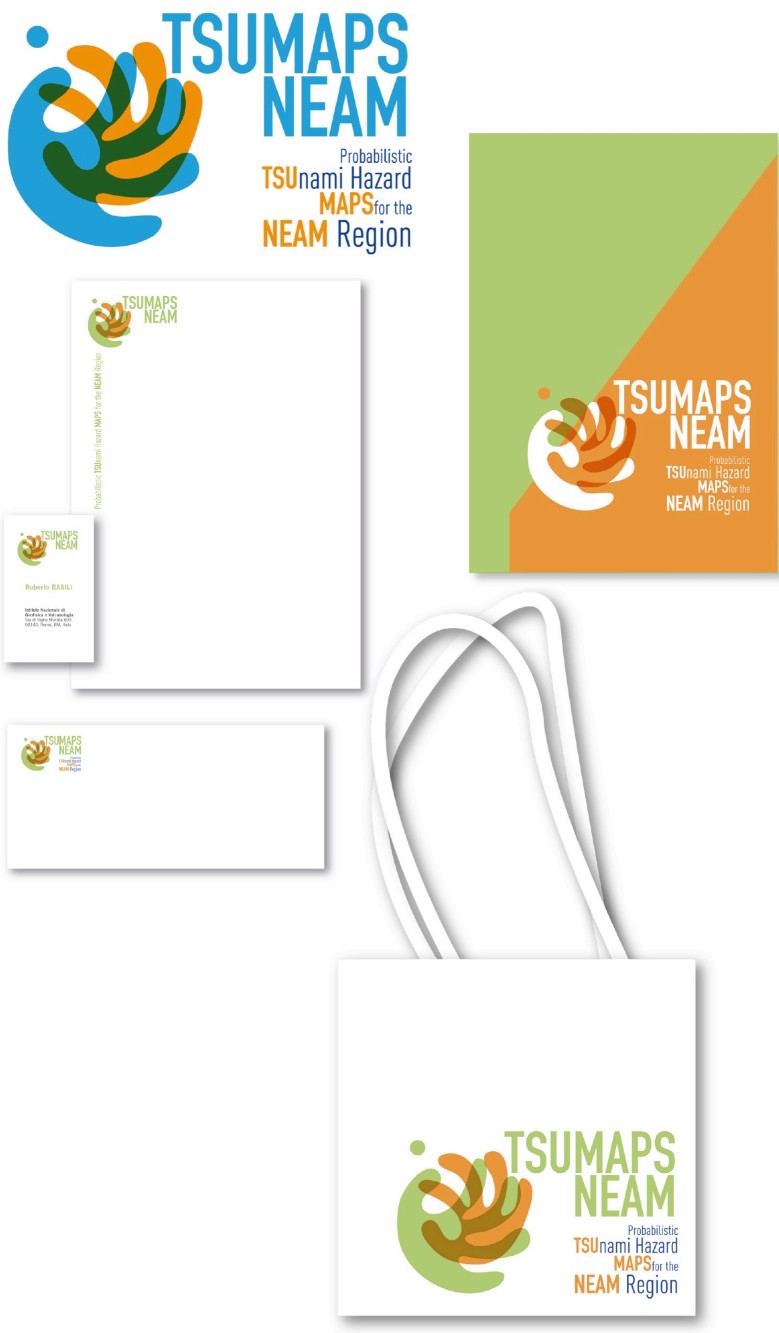

**Figure 5: The TSUMAPS NEAM Project coordinated image. Copyright Laboratorio Grafica e
Immagini INGV.**





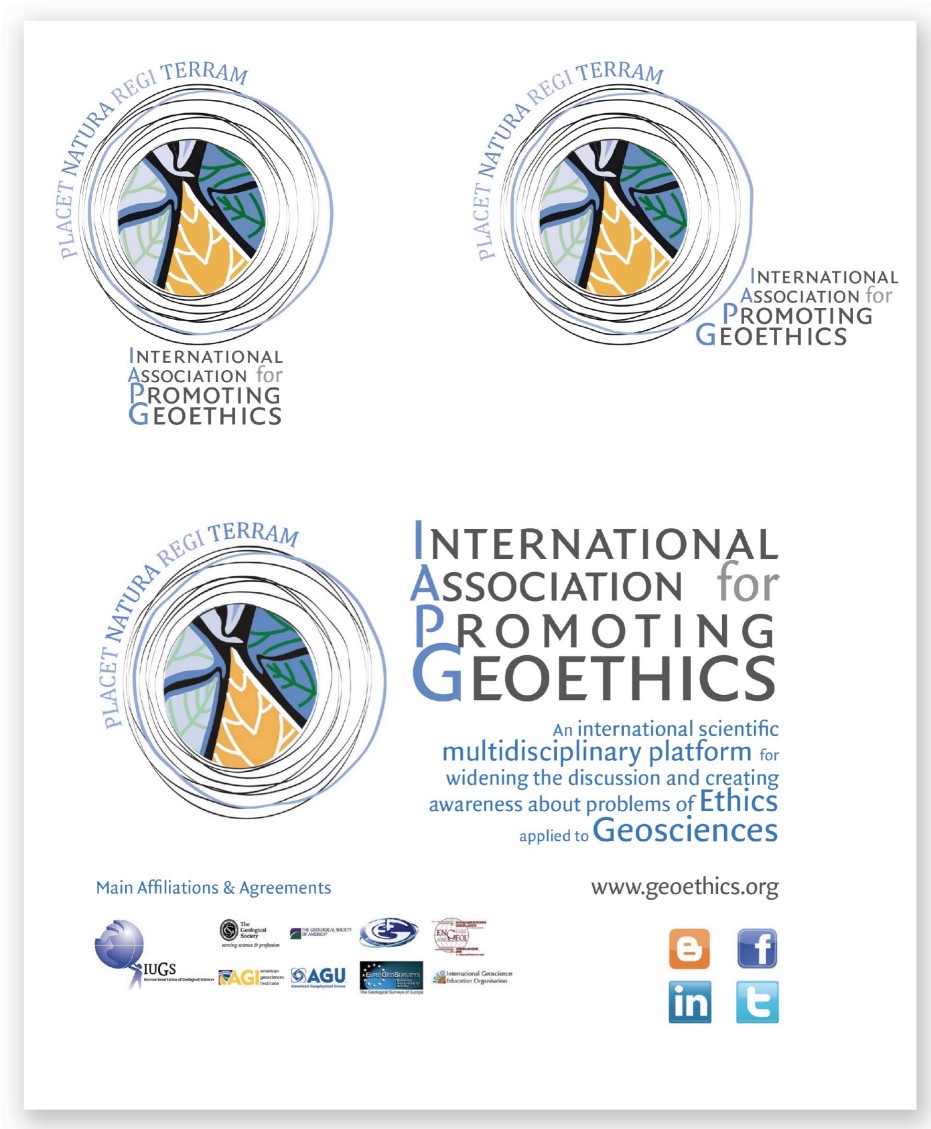

**Figure 6: Some achievements for the IAPG. Copyright Laboratorio Grafica e Immagini INGV.**






**Figure 7: The solution chosen for AGITHAR and some gadgets. Copyright Laboratorio Grafica e Immagini INGV.**




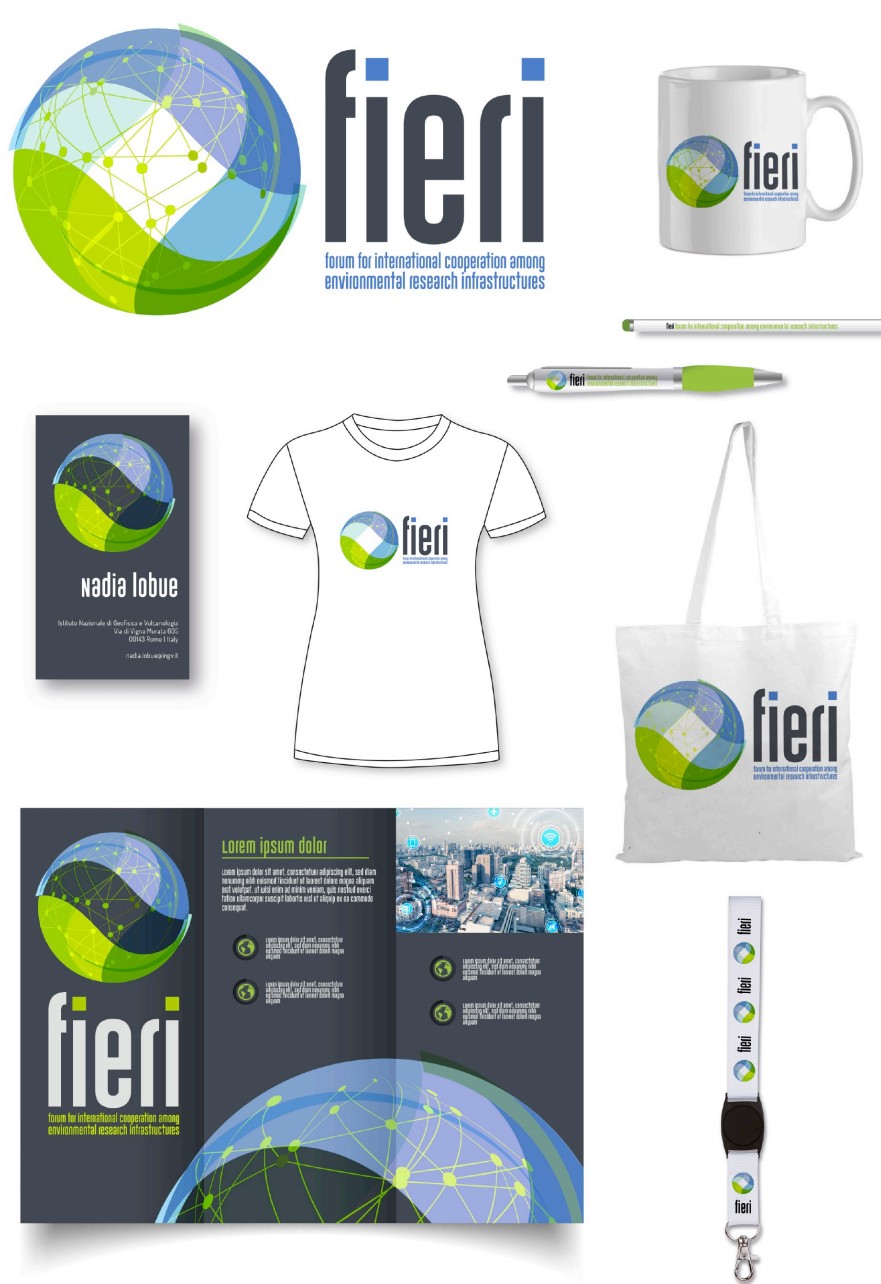

**Figure 8: The FIERI solution. Copyright Laboratorio Grafica e Immagini INGV.**









**Figure 9: Savemedcoasts products. Copyright Laboratorio Grafica e Immagini INGV.**




Figure 10: ScienzaInsieme products. Copyright Laboratorio Grafica e Immagini INGV.






**Figure 11: The last issues of Annual Report of the HP-HT Laboratory. Copyright Laboratorio Grafica e Immagini INGV.**






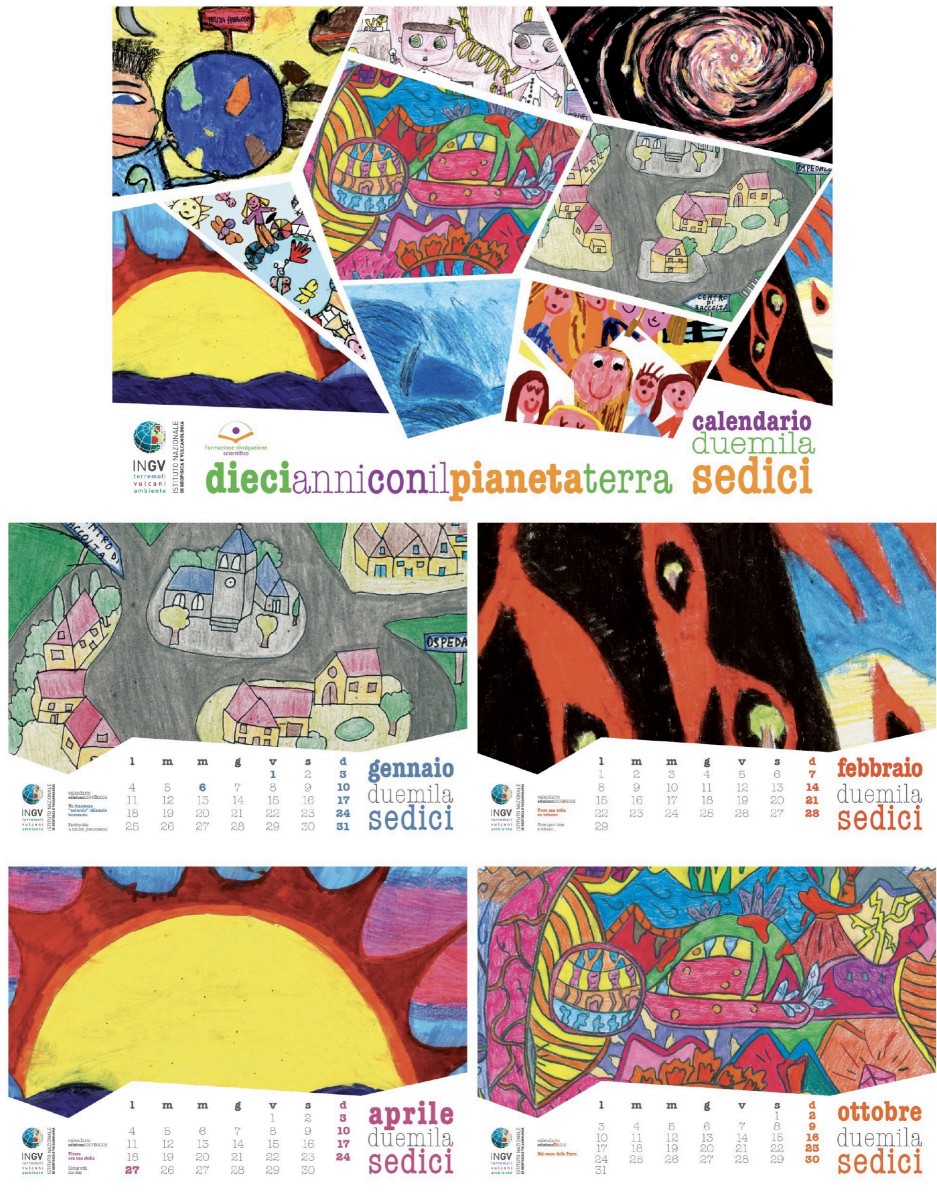

**Figure 12: Covers and declinations on the 2016 calendar's monthly agenda. Copyright Laboratorio Grafica e Immagini INGV.**








Figure 13: The billboard of Escape Volcano game. Copyright Laboratorio Grafica e Immagini INGV.






**Figure 14: Mareopoli game. The game board, the playing cards, the dice and one of the illustrations created by the Laboratory. Copyright Laboratorio Grafica e Immagini INGV.**




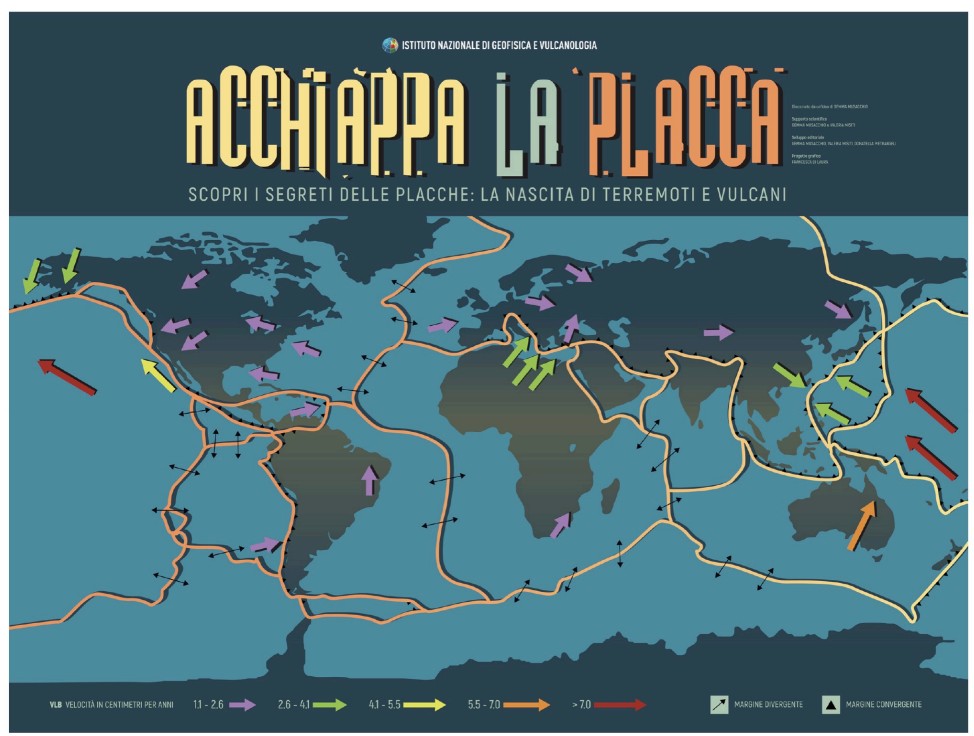

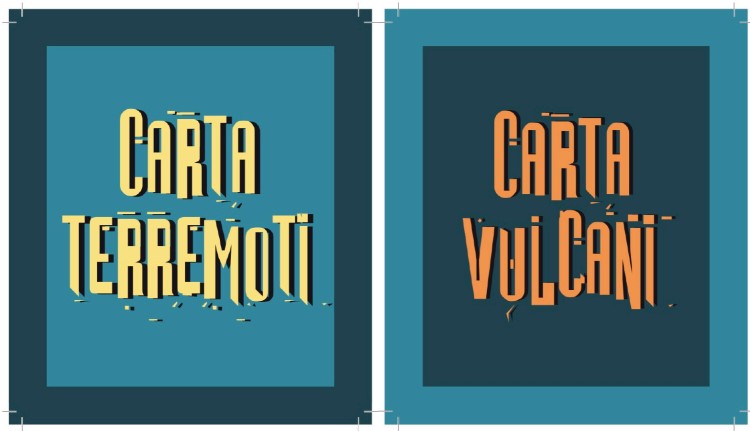

**Figure 15: The billboard and the cards of Catch the plate Game. Copyright Laboratorio Grafica e Immagini INGV.**






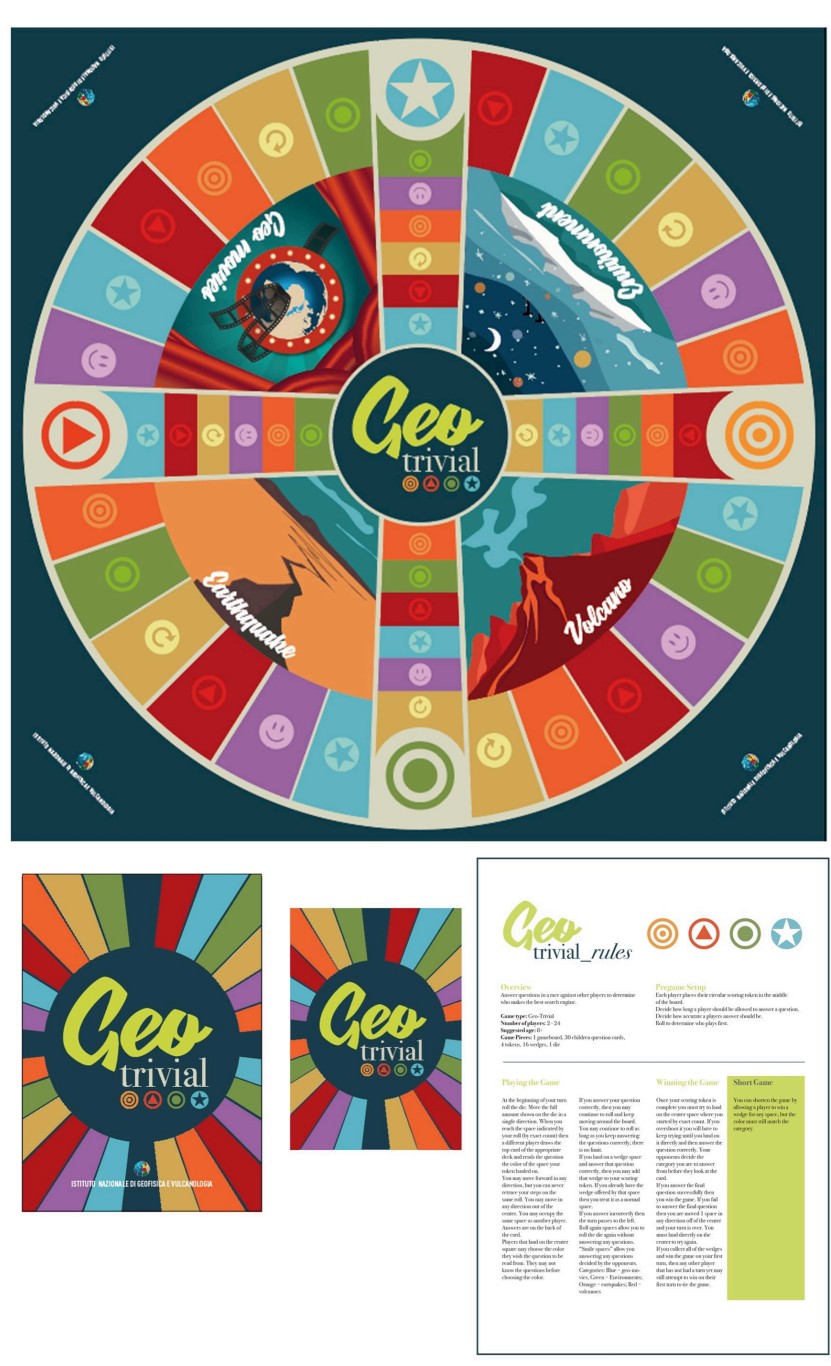

**Figure 16: The GEO Trivial Game, the board, the cards and the rules. Copyright Laboratorio Grafica e Immagini INGV.**




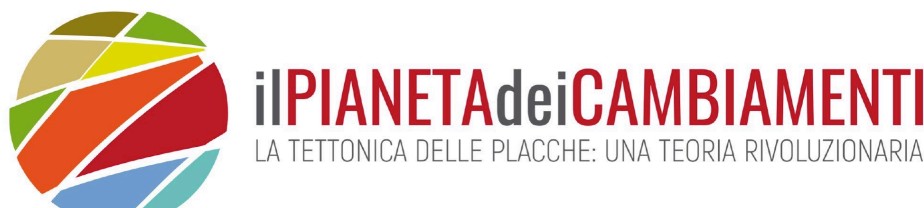

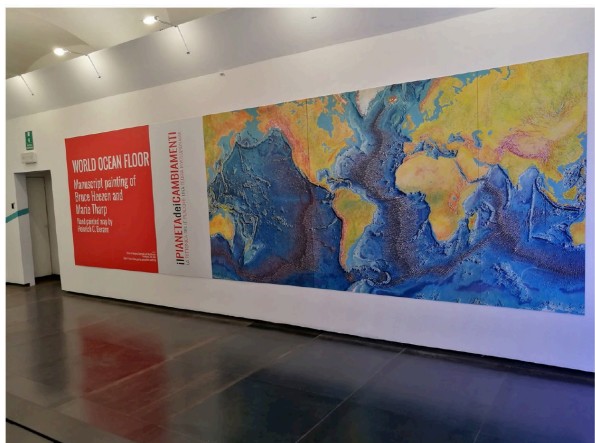

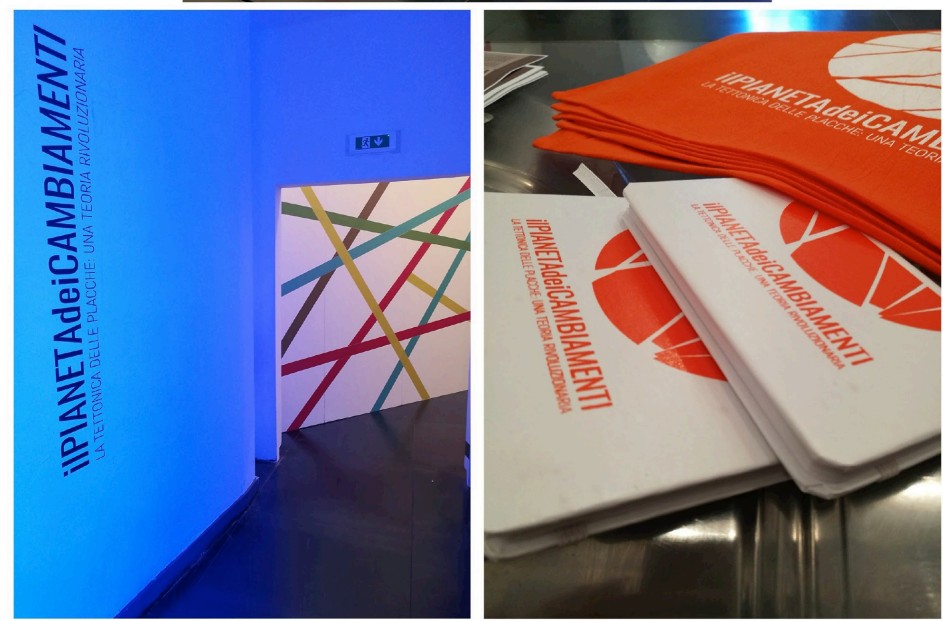

**Figure 17: Some details of the exhibition layout Il pianeta dei cambiamenti. La tettonica delle placche: una teoria rivoluzionaria - Festival della Scienza di Genova 2018, at Palazzo Ducale (Genova, Italy). Copyright Laboratorio Grafica e Immagini INGV.**







**Figure 18: Some pages of the Exhibition Catalog: Il pianeta dei cambiamenti. La tettonica delle placche, una teoria rivoluzionaria. Copyright Laboratorio Grafica e Immagini INGV.**





**Figure 19: The exhibition at the 2019 Edition Science Festival (Commenda da Prè, Genova, Italy).**







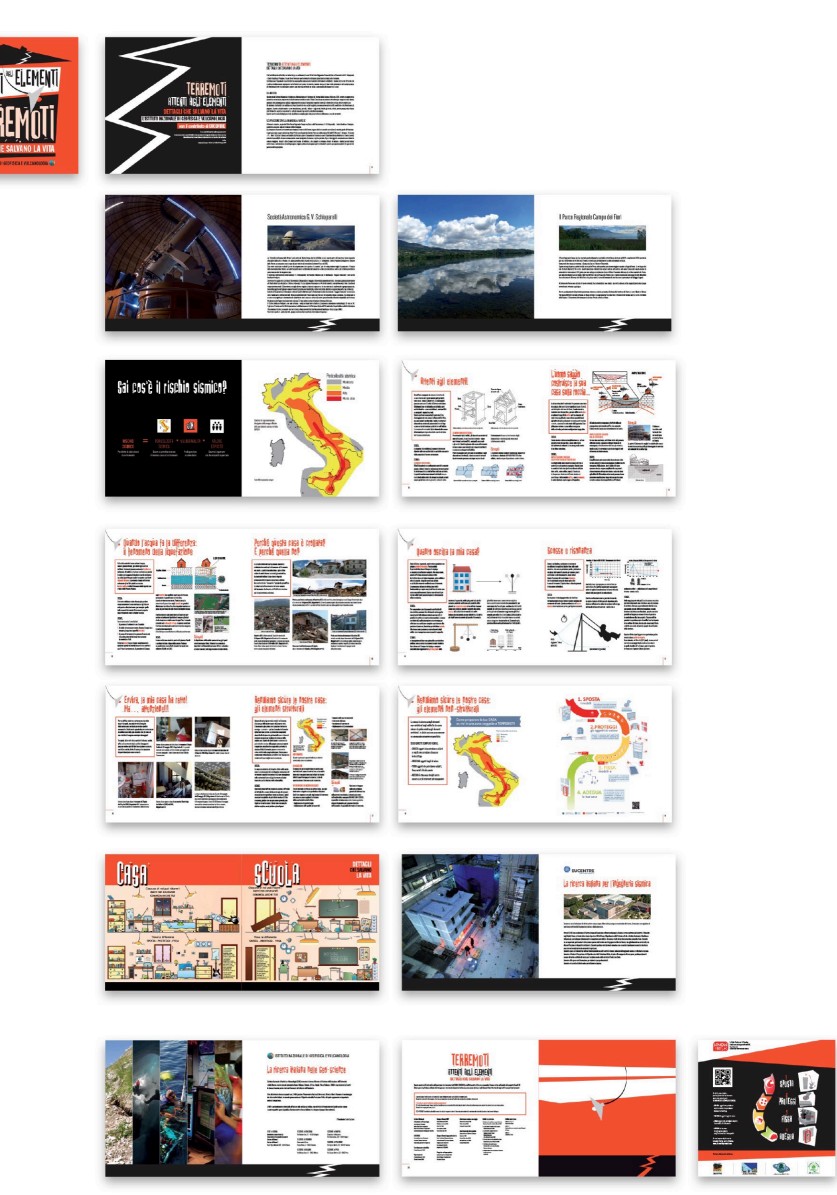

**Figure 20: The booklet of the exhibition: Attenti agli elementi. Copyright Laboratorio Grafica e Immagini INGV.**





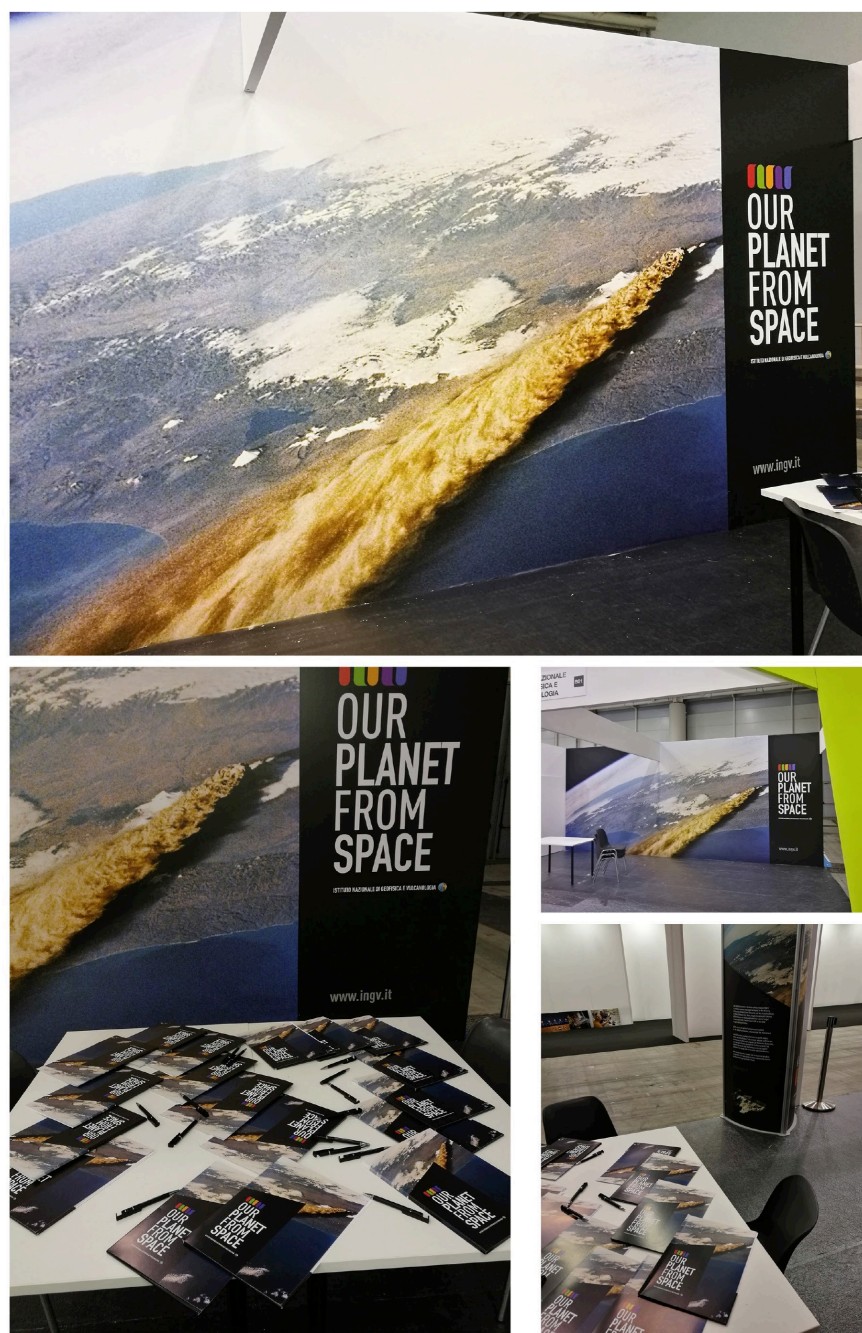

**Figura 21: The INGV participation at the NSE Forum, December 2019, Rome, Italy. Copyright Laboratorio Grafica e Immagini INGV.**







**Figura 22: The production for the INGV participation at the NSE Forum, December 2019, Rome, Italy. Copyright Laboratorio Grafica e Immagini INGV.**
