# Peer review of "Graphic design and scientific research: the INGV experience"

_Geoscience Communication, 2020_

## Referee Comment (RC1) · Jacco Konijn (Referee) · 25 May 2020

Both abstract, introduction and conclusions should more clearly express the fact that this is written out of personal experience within INGV. There have not been any studies evaluating and analysing the effectivity of the graphical design of the activity, organisation, project or tool that is presented in the article. So any conclusion is personal experience only.

Present the article more clearly as an overview article of graphical design for dissemination activities within INGV. Dissemination responsible persons could benefit from this article, but this target audience should be addressed more directly.

It is highly recommended that the article is reviewed by an English native speaker on

language, grammar and spelling, before publishing it for an international audience.

---

## Referee Comment (RC2) · Anonymous Referee #2 · 26 May 2020

I read this paper with interest, as I consider the developments it describes really important for effective communication in science. I have no doubt that the paper and its beautiful iconography will definitely be of inspiration for many.

Unfortunately, the quality of the text does not match the value of the work presented. I see three main limitations.

First of all, the introductory parts do no explain clearly what the reader will find in the text. The paper is interesting and rich, but the experiences it describes appear one on top of the other without any prior guidance. The authors would have gotten much better recognition for their work if they identified and clearly listed in some orderly fashion the different areas of their activity: corporate, large project, participation to meetings, edugames etc.

[Figure]

The second major obstacle is the English, that is poor and definitely needs a thorough review by a professional translator.

Due to the poor quality of the translation, the logic implied in some statements is sometimes flawed, and this is a further limitation. This implies that the translator must be carefully followed to make sure that he/she understands exactly what the authors meant to say.

I truly hope that the authors will do their best to improve the text a reasonable publication standard. As I alredy stated, the quality of the illustrations is very good, and the display of the Laboratorio's past activity really outstanding. Some of the drawings are truly awesome small pieces of art (e.g that of the Tsumaps-NEAM project) that make some logos of just a few decades ago appear as pre-history.

---

## Author Comment (AC1) · 20 Jul 2020

Dear Editor and Reviewer1, Please find below our point by point response (A) to the reviewers' comments (R1).

R1. Both abstract, introduction and conclusions should more clearly express the fact that this is written out of personal experience within INGV.

A. In the abstract, introduction and conclusion we have underlined that what we described is based on INGV's team experiences.

R1. There have not been any studies evaluating and analysing the effectivity of the graphical design of the activity, organisation, project or tool that is presented in the article. So any conclusion is personal experience only.

[Figure]

A. There are no specific studies on the effectivity of the graphical design in the INGV activity as this was the first synthesis that relate and describe the interaction and the synergy between graphic designers and researchers in a common research work. When considering aesthetic sensitivity, quantitative analysis is difficult to estimate. However, we have added the number of visitors and the positive evaluations on graphic framework recorded in the visitor's guestbooks during INGV temporary exhibitions, the number of participants in the educational games and their comments.

R1. Resent the article more clearly as an overview article of graphical design for dissemination activities within INGV. Dissemination responsible persons could benefit from this article, but this target audience should be addressed more directly.

A. The introduction has been modified to clarify the context and purpose of this paper, also with the aim to better steer readers.

R1. It is highly recommended that the article is reviewed by an English native speaker on language, grammar and spelling, before publishing it for an international audience.

A. The manuscript is under revision for English language, grammar and spelling.

---

## Author Comment (AC2) · 20 Jul 2020

Dear Editor and Reviewer2, Please find below our point by point response (A) to the reviewers' comments (R2).

R2. . . .the introductory parts do no explain clearly what the reader will find in the text. The paper is interesting and rich, but the experiences it describes appear one on top of the other without any prior guidance. The authors would have gotten much better recognition for their work if they identified and clearly listed in some orderly fashion the different areas of their activity: corporate, large project, participation to meetings, edu-games etc.

A. The introduction has been modified to clarify the context and purpose of this paper,

also with the aim to better steer readers. We have listed homogeneously the experiences giving a clearer order to the activities.

R2. The second major obstacle is the English, that is poor and definitely needs a thorough review by a professional translator.

A. The manuscript is under revision for English language, grammar and spelling.